# Selenium Modification of Natural Products and Its Research Progress

**DOI:** 10.3390/foods12203773

**Published:** 2023-10-13

**Authors:** Kaixuan Cheng, Yang Sun, Bowen Liu, Jiajia Ming, Lulu Wang, Chenfeng Xu, Yuanyuan Xiao, Chi Zhang, Longchen Shang

**Affiliations:** 1College of Biological and Food Engineering, Hubei Minzu University, Enshi 445000, China; 202230332@hbmzu.edu.cn (K.C.); a13842765391@icloud.com (Y.S.); lbw3.1415926@gmail.com (B.L.); 202130352@hbmzu.edu.cn (L.W.); 17856887693@163.com (C.X.); 202130350@hbmzu.edu.cn (Y.X.); zhtzu@163.com (C.Z.); 2Enshi Tujia and Miao Autonomous Prefecture Academy of Agricultural Sciences, Enshi 445000, China; jiajiaming77@163.com

**Keywords:** natural products, selenium, chemical modification, biological activity

## Abstract

The selenization of natural products refers to the chemical modification method of artificially introducing selenium atoms into natural products to interact with the functional groups in the target molecule to form selenides. Nowadays, even though scientists in fields involving organic selenium compounds have achieved numerous results due to their continuous investment, few comprehensive and systematic summaries relating to their research results can be found. The present paper summarizes the selenization modification methods of several kinds of important natural products, such as polysaccharides, proteins/polypeptides, polyphenols, lipids, and cyclic compounds, as well as the basic principles or mechanisms of the selenizing methods. On this basis, this paper explored the future development trend of the research field relating to selenized natural products, and it is hoped to provide some suggestions for directional selenization modification and the application of natural active ingredients.

## 1. Introduction

Selenium (Se) is an essential trace element with diverse biological functions within the body [1]. An insufficient intake of selenium-containing foods can lead to various diseases related to selenium deficiency [2], such as Keshan disease [3], heart disease [4], tumors [5], and immune injury [6]. According to reports, about two-thirds of the world’s soil is selenium-deficient, and there are about 5 million to 1 billion people who are facing the threat of selenium deficiency-related diseases [7]. Specifically, Central China and southeastern Siberia are highly insufficient in selenium [8]. In China, nearly 72% of the area is insufficient in selenium to various degrees, and there are as many as tens of millions of people who are insufficient in selenium [7]. The selenium level of the selenium-deficient regions is significantly lower than that of the selenium-rich areas, resulting in 39-61% of Chinese residents having a daily selenium intake under the recommended value of the World Health Organization/FAO [9]. Increasing dietary selenium intake can effectively reduce the risk of these diseases [10].

Organic selenium occurs in nature mainly in the form of selenized polysaccharides and selenized polypeptides, among others. While inorganic selenium compounds mainly include selenates, selenites, etc. [11]. Compared with inorganic selenium, organic selenium has significant advantages such as high bioavailability and low toxicity, as well as it being more easily absorbed by the human body [12]. Selenization modification is an effective method to convert inorganic selenium into organic selenium artificially. During selenization modification, the selenium element of inorganic selenium compounds binds to organic compounds such as natural products and interacts with their functional groups, thereby introducing selenium into organic compounds to achieve the selenization modification of target compounds, changing their properties and functions [12] and further tapping its application potential. The selenization modification of natural products involves many fields, such as biochemistry, molecular biology, pharmacy, and food science [1]. As a newly developed chemical modification method, selenization has a brief research history [13]. However, due to the unique physiological activity of selenized products, exploring how to efficiently selenize them and delve deeper into their physiological activity has become one of the research hotspots in recent years. At present, various selenylation modification methods have been widely used to improve the biological activity of natural products and develop new selenium supplements [14]. This paper reviews the methods and fundamental principles used for selenization modification of significant natural products such as polysaccharides, proteins, and polyphenols. The operating parameters and binding modes involved in selenide modification are also summarized. On this basis, the biological activity and application of selenized derivatives of natural products are discussed, and their future development prospects are presented. It is expected to provide some theoretical reference for the development and application of natural product selenide modification technology.

## 2. Absorption and Metabolism of Selenium

The inorganic selenium in soil could be absorbed by plants to form an organic one, which is then transmitted to animals through the food chain [15]. Ingestion of the food of a plant or animal origin containing selenium was the primary way for mammals to absorb selenium. In mammal organisms, selenium mainly exists in the form of sel-enate, selenite, selenomethionine (SeMet), and selenocysteine (SeCys) [16]. The ingested selenium compounds with a large molecular weight, such as selenium-containing proteins, could be absorbed and utilized after hydrolyzing into small-molecular-weight selenium compounds [17]. Among them, the SeMet and SeCys could be effectively absorbed through the transmembrane pathway mediated by amino acid transporters in their intestinal tract and then participate in the metabolic process in vivo. In comparison, selenate and selenite are usually absorbed through passive diffusion and co-transport mechanisms [7].

The absorbed selenium in mammals was distributed in the two metabolic pools of their body. In the SeMet metabolic pool, the SeMet can non-specifically and randomly replace methionine to be incorporated into proteins and can also be converted into SeCys through the trans-sulphuration pathway. In the Se regulatory metabolic pool, different Se sources are converted into hydrogen selenide (HSe^−^), such as selenite, which can react with reduced glutathione to form selenodiglutathione, and finally convert into hydrogen selenide HSe^−^ for the synthesis of selenoprotein, which is then distributed to various organs and tissues. The excess hydrogen selenide could also be metabolized in the methylation pathway and be excreted from urine as soluble trimethyl selenium ions or selenosugars or be excreted by respiration in the form of volatilization of dimethyl selenide [17,18].

## 3. Selenium Modification of Polysaccharides

### 3.1. Biological Activity of Selenium Polysaccharides

Even in a selenium-rich environment, the content of natural selenium polysaccharides in organisms is still low [19], which makes it difficult to support the increasing demand for selenium polysaccharide production. Therefore, more effective measures are needed to improve the efficiency of polysaccharide selenization. Combining natural polysaccharides with inorganic selenium compounds not only overcomes the shortcomings of natural selenium polysaccharides with few sources and low selenium content, but also enhances the target products by incorporating the biological activities, such as antioxidant, anti-tumor, immunity enhancement, liver protection, anti-diabetes, anti-inflammation, and neuroprotection, of both polysaccharides and selenium elements [20]; the activity of the resulting products is often far superior to the activity of selenium or polysaccharides by themselves [21].

### 3.2. Polysaccharide Selenization Pathway

Common polysaccharide selenization modification methods mainly include the following methods [22]. One is the selenization modification of polysaccharides with selenium, selenite, or selenate under mild conditions. Due to the low selenization efficiency of this method, its application is not extensive and there is little related literature. The second is the selenium oxychloride (SOC) selenization method, which is a commonly used method for polysaccharide selenization modification. Polysaccharide molecules contain a large number of hydroxyl groups, and chlorine selenate (SeClO_2_) has active acyl chloride groups. As an active acylating agent, it can form an ester bond with the hydroxyl group in the polysaccharide [23], thereby connecting selenium and polysaccharides to complete the selenization modification of polysaccharides [24]. However, the synthesis conditions of SeClO_2_ are harsh and the cost is high. As an acyl chloride reagent, SeClO_2_ reacts more strongly with polysaccharides, posing potential safety risks in the experiments [11]. Thus, the selenium functional group method [14] has been an essential advancement in the selenium modification of polysaccharides. The main objective of this method is to create a connection between selenium and polysaccharide molecules by using selenium-containing functional groups, thereby achieving the selenization of polysaccharides. Common selenium functional group methods include the nitric acid–sodium selenite method and the acidic ionic liquid–selenous acid method, among others.

The selenization modification methods of polysaccharides are summarized in Table 1. There are many methods for selenization modification by the selenium functional group method to obtain selenized polysaccharides, including aluminum chloride–sodium selenite (AlCl_3_-Na_2_SeO_3_), nitric acid–sodium selenite (HNO_3_-Na_2_SeO_3_), nitric acid–selenous acid (HNO_3_-H_2_SeO_3_), glacial acetic acid–selenous acid (C_2_H_4_O_2_-H_2_SeO_3_), and glacial acetic acid–sodium selenite (C_2_H_4_O_2_-Na_2_SeO_3_). Although various selenization methods can be used to modify polysaccharide molecules, the HNO_3_-Na_2_SeO_3_ method is favored for its superior operability, good safety, and high selenization efficiency [11]. The general process route of the method is shown in Figure 1. Usually, barium chloride and sodium selenite are added after the polysaccharides are dissolved in nitric acid. Among them, HNO_3_ can create an acidic environment for the reaction system so that the active groups on the polysaccharide chain react with the selenizing reagent. BaCl_2_ has strong coordination with the hydroxyl group, which can enhance the nucleophilicity of the hydroxyl group and is conducive to the formation of a stable single complex [25]. After the reaction, the pH is adjusted with a saturated sodium carbonate solution. Then, an appropriate amount of Na_2_SO_4_ solution is added to form a precipitate to remove Ba^2+^ ions. The supernatant is dialyzed (low molecular weight cut-off) and lyophilized to obtain selenized polysaccharides [26]. Hu et al. [27] used the HNO_3_-Na_2_SeO_3_ method to modify *mulberry leaf* polysaccharides (MLPs) and found that the antioxidant capacity of selenized MLPs was improved (Figure 2a). Li et al. [28] prepared Se-DDP by the selenization of *Dendrobium Devonianum Paxt* (DDP) polysaccharides with sodium selenite as the selenization agent and glacial acetic acid as the catalyst. The Se-DDP sample of 8 mg/mL showed an antioxidant capacity not inferior to that of vitamin C.

In addition, it is not difficult to see from Table 1 that a variety of polysaccharides can be successfully selenized by hydrothermal synthesis with the HNO_3_-Na_2_SeO_3_ system. However, the reaction time is too long (about 5–10 h) and the acidity of nitric acid is strong, resulting in greater acid contamination [29]. Moreover, the polysaccharide chain structure is also easily degraded during the hydrothermal process [37]. The application of the AlCl_3_-Na_2_SeO_3_ system can overcome the above shortcomings [29]. At the same time, ultrasound, microwave, vacuum, alkali treatment, enzymatic hydrolysis [10,21,38], and other auxiliary methods can be used to improve selenization efficiency (saving time and increasing yield). Yue et al. [37] used the optimized microwave-assisted method to prepare selenylation Astragalus polysaccharides with the HNO_3_-Na_2_SeO_3_ system. It was found that microwave treatment could improve the synthesis efficiency of polysaccharide derivatives and polysaccharide-based copolymers. Selenium was found to form selenium ester bonds (O-Se-O) and hydrogen bonds (O-H⋅⋅⋅Se) with astragalus polysaccharide chains. Additionally, in a particular acidic environment with specific conditions, inorganic selenium can also combine with hydroxyl (−OH), amino (−NH_2_), aldehyde (-CHO), carbonyl (−C=O), and other groups on the polysaccharide sugar chain in the form of covalent bonds to form polysaccharide selenite or polysaccharide selenate [25]. As shown in Figure 2, selenium was combined with the hemiacetal hydroxyl group at the C-6 position of the polysaccharide residue in the form of a covalent bond to form selenate. The characteristic absorption peaks of Se-O-C, Se=O, or O-Se-O are usually observed at about 670, 760–860, and 1030 cm^−1^ on the infrared spectra of selenized polysaccharides [39]. In the nuclear magnetic resonance (NMR) spectrum, a new C-6 signal peak appears at about δ62.9 [40]. Yang et al. [36] used acidic ionic liquids as solvents and catalysts to activate selenous acid for the selenium modification of polysaccharides, which overcame the poor solubility of polysaccharides in many non-protonic solvents. Among them, selenium and polysaccharides were also connected by Se-O-C and Se=O. The inhibitory effects of three selenized polysaccharides on human liver cancer cells (Hep G2) and human colon cancer cells (COLO205) were significantly improved where the selenium content of the polysaccharides had a certain effect on their inhibitory effects. The selenium modification scheme is shown in Figure 2.

### 3.3. Selenium Content of Selenopolysaccharides

The selenium content largely affects the biological activity of selenopolysaccharides [33], and the selenium content of selenized polysaccharides is mainly determined by the type of polysaccharides and the method of selenization. Li et al. [41] used the HNO_3_-Na_2_SeO_3_ method to selenize Epimedium polysaccharides and Radix Isatidis polysaccharides with nine reaction conditions. It was observed that the selenium content of Epimedium polysaccharides and Radix Isatidis polysaccharides is different under the same operating parameters. For example, under the same operating parameters, that is, 300 mg sodium selenite reacted at 90 °C for 10 h, the selenium content of selenized Epimedium polysaccharides (19.32 mg/g) was greater than that of selenized Radix Isatidis polysaccharides (12.87 mg/g). Gao et al. [31] used the HNO_3_-Na_2_SeO_3_, C_2_H_4_O_2_-H_2_SeO_3_, C_2_H_4_O_2_-Na_2_SeO_3_, and SOC methods to selenize *garlic* polysaccharides. It was found that the selenium content of selenized polysaccharides by different methods was quite different, and the selenium yield could be approximately compared as HNO_3_-Na_2_SeO_3_ > C_2_H_4_O_2_-H_2_SeO_3_ > SOC > C_2_H_4_O_2_-Na_2_SeO_3_. Mangiavacchi et al. [42] explored the modification of cyclic and polyhydroxylated chains of selenosugars and studied their inhibitory activities on glycosidase. The substitution of sulfur with selenium in those molecular compounds positively affected inhibitory activity, and some of the selenosugars showed selective inhibition of specific glycosidases. Their research results revealed that the stereochemistry and cyclic configuration of substituents are critical factors affecting the activities of these selenium-modified polysaccharides. In the selenium modification of polysaccharides, different selenium modification methods should be selected according to the actual situation for different types of polysaccharides. The focus is on regulating the type of selenium source, the amount of selenium, and the reaction parameters to achieve the best selenization effect so that the selenized polysaccharides can exert the best functional properties.

## 4. Selenium Modification of Protein/Polypeptides

Proteins are larger biomolecules composed of one or more polypeptide chains. Polypeptides are biomaterials composed of repeating amino acid units linked by a peptide bond. Selenium-containing proteins and selenium-containing polypeptides are the main forms of selenium in animals and plants, and they are also an important material carrier for selenium to exert its physiological activity [43].

### 4.1. Protein Selenization Modification

Protein selenization modification refers to the introduction of one or more selenium atoms into proteins. This modification is usually achieved by the reaction of proteins with compounds such as selenates. Selenium modification contributes to protein structure stability and enhances biological activity [44]. Zhang et al. [45] used Na_2_SeO_3_ as the selenium source to prepare selenized Morchella protein hydrolysate by the microwave-assisted method. The selenium content reached 59.0 mg/g, with obvious cell protection and excellent safety performance.

Researchers usually optimize the selenization process by changing the selenium source and modification method. The common protein selenization modification methods are shown in Table 2. Zhao et al. [46] modified selenium proteins with dry heat treatment in the presence of selenite, which increased the selenium content of egg white proteins, enhanced the antioxidant properties of the proteins, and prolonged their shelf life. This selenization modification method has the advantages of short time, low cost, and a high selenization level. Yu et al. [47] explored the anti-tumor effect of selenized β-lactoglobulin (Se-β-Lg) on human gastric cancer cells. They found that Se-β-Lg significantly inhibited the proliferation of human gastric cancer cells with typical apoptotic morphological characteristics by inducing G2/M phase cell cycle arrest. Zhao et al. [46] found that in the process of protein selenization, selenium combined with the hydroxyl groups of amino acids to form a selenous ester bond (-O-SeHO_2_), and heating promoted the formation of the selenous ester bond. The reaction process is shown in Figure 3, where Pro represents the protein. In addition, Zhang et al. [48] modified the protein hydrolysate of Morchella esculenta by selenization. They found that the methionine residues amino, hydroxyl, and sulfur atoms were involved in the selenization reaction. The thiol group (-SH) containing sulfur atoms also participates in the selenization reaction. Among them, thiols such as cysteine react with selenite as follows: 4RSH + H_2_SeO_3_ → RSSR + RSSeSR + 2H_2_O [49], where R represents the cysteine group (−CH_2_CH(NH_2_)COOH).

### 4.2. Selenium Modification of Polypeptides

Selenium modification of peptides refers to the reaction of exogenous selenium with groups of peptides, thereby introducing selenium atoms into the target peptide to achieve selenium modification of the peptide molecule. Selenopeptides have been shown to have various biological effects such as anti-tumor, neuroprotection, antibacterial, anti-toxic heparin, and immunomodulation [3]. Polypeptides are more easily absorbed by the human body than proteins, so selenylation of peptides may be more important for promoting human health [6]. Selenium and sulfur have similar chemical properties; but, compared with sulfur, selenium has lower electronegativity and a larger atomic radius [52]. Therefore, the Se-H bond is easier to dissociate and oxidize than the S-H bond, resulting in a higher acidity of selenocysteine than cysteine. These differences can be used to selectively modify peptides by introducing new functional groups, which is of great significance for the folding and stability of protein molecules [53].

Selenized polypeptides can be obtained by biosynthesis and by artificial synthesis. At present, a variety of selenization modification methods have been used for the selenization of peptides. These methods include microbial preparation, chelation, gene recombination, and the enzymatic method [54], among which the chelation method is the most commonly used chemical method for artificial direct selenization of polypeptides. Chelation usually uses selenite as the selenium source, and the operating parameters of this method are shown in Table 3.

Chelation refers to the addition of selenium to natural peptides and the synthesis of selenized peptides by chelation. According to the molecular orbital theory, O and N can provide lone pair electrons to form coordination bonds [57]. The chelating sites of selenium and peptides are usually around O and N. Se^4+^ can provide 4d empty orbits, while N and O can provide lone pairs of electrons to occupy empty orbits, thus forming coordination bonds between selenium and these atoms. Li et al. [56] modified anticancer bioactive peptides (ACBPs) with S-acetyl mercapto succinic anhydride by chemical absorption, in which the binding site of Se was linked with the sulfhydryl group, which indicated that the ACBP chain could be linked with the sulfhydryl group through an amide bond to form an ACBP-chelated selenium complex (Figure 4). While Ye et al. [6] found that an extension of the chelation reaction time will lead to the decomposition of soybean protein isolate polypeptide selenium and the temperature and pH value will also affect the chelation.

The use of selenide peptides in functional foods can promote the bioavailability and stability of micronutrients [5]. Doan et al. [58] used sodium selenite, soybean protein-chelated selenium, and selenium yeast to feed weaned piglets intraperitoneally, and found that soybean protein-chelated selenium is a highly effective organic selenium supplement, which can significantly reduce oxidative stress and cell damage. Selenium-chelated peptides are less toxic than inorganic selenium, have high safety and strong antioxidant capacity, and can be used as a high-efficiency organic selenium supplement in food or pharmaceutical fields in the future.

## 5. Selenium Modification of Polyphenols

Polyphenols are a kind of bioactive substance that naturally exists in animals and plants, which has many physiological functions such as antioxidant, anti-inflammatory, and anti-tumor [59]. Polyphenols can be connected with selenium through their phenolic hydroxyl groups. When +4-valent Se coordinates with the polyphenol oxygen anion, the d orbital of Se can form a common electron pair with the p orbital of the polyphenol oxygen anion, thus forming a covalent bond [60]. Studies have found that selenium and polyphenols have synergistic effects. Fiorito et al. [61] used the rich content of auraptene in citrus fruits for selenization modification and found that its selenium-containing derivatives had stronger in vitro antioxidant capacity and free radical scavenging activity than auraptene itself (Figure 5).

Currently, regarding the selenide modification of polyphenols, most of the relevant studies are on flavonoids [62], as shown in Table 4. Zhu et al. [63] realized the construction of C-Se bonds only by C-H functionalization at the α-position on the flavonoid skeleton using Cu as a catalyst. The method uses cheap and easily available potassium selenocyanate as a selenium source. It was confirmed by NMR spectroscopy that the aryl selenium (ArSe-) group selectively replaced the α-position of flavone ketone functions to generate flavonoid selenides, and the yield was good. Logheswaran et al. [64] prepared selenoflavone by a ruthenium-catalyzed selenylation reaction, using simple unsaturated acids/amides and diaryl selenium to generate various alkenyl selenides with different degrees of substitution. A variety of diaryl selenium compounds can be used as starting materials for this method to generate selenium flavonoids with different substituents and certain medicinal value, and the method has excellent product stereoselectivity and reaction site selectivity. Choi et al. [65] used bromobenzene and cinnamoyl chloride to treat with tert-butyl lithium in the presence of AlCl_3_ and then reacted this with elemental selenium to obtain selenoflavanone by two-step synthesis. The total yield reached 76–80%, and selenoflavanone showed lower polarity and higher lipophilicity than flavanone, indicating that it was easier to penetrate the blood–brain barrier. In addition, the study confirmed that the total infarct volume of selenoflavanone was significantly reduced when used to treat mice with transient cerebral ischemia, showing stronger neuroprotective activity than flavanone. Figure 6 shows the selenization process of the polyphenols mentioned above. Marć et al. [62] evaluated the antimicrobial activity of selenoflavones and their bioisosteric analogues. They found that these compounds showed concentration-dependent antibacterial activity against both the Gram-positive and Gram-negative bacteria. Among them, at a concentration of 200 μM, selenoflavone showed the highest antimicrobial activity against Botox, whose growth was inhibited by 57%. In contrast, the growth of E. coli was inhibited by 49%. However, none of the compounds showed antimicrobial activity against the yeast cells, even at a higher concentration.

To summarize, selenium-modified polyphenols have a noteworthy effect on enhancing animal growth performance, antioxidant function, and immune function. Most studies indicate that the physiological activity of selenium-modified polyphenols surpasses that of selenium or polyphenols alone. Nevertheless, there is still a research gap on the reaction principle, structure–activity relationship, and physiological activity of selenium-modified polyphenols which researchers need to explore further.

## 6. Other Selenization Modification

### 6.1. Selenium-Modified Cyclic Compounds

Cyclic selenide and diselenide drugs have antioxidant, redox regulation, anticancer, and radioprotective activities [69], which have a potential application value in cancer prevention and treatment. The C-H bond is the most common structural fragment in organic compounds. Converting inert C-H bonds into C-Se bonds avoids pre-functionalization steps, thus simplifying the synthesis steps and reducing the generation of by-products, which can achieve a higher atomic economy [70]. Therefore, how to efficiently form selective C-Se bonds is crucial for the synthesis of cyclic selenides. The use of transition metals to catalyze the cross-coupling of boronic acid aryl/halogenated aromatics with selenium sources is a common method for selenizing cyclic compounds [71]. Cross-coupling reactions usually use transition metals such as ruthenium, silver, and copper as catalysts. Figure 7 shows common biologically active aryl/heterocyclic aryl selenides. For example, Jakubczyk et al. [72] used Cu as a catalyst, N-substituted heteroaromatics (such as 4-nitro pyrazole or 4-nitroimidazole) as the starting material, and aryl iodides (iodinated benzene) as the source of aryl groups in the reaction, and used elemental selenium to obtain asymmetric cyclic aryl selenides containing multiple substituents (Figure 8). As a result, the C-H bond in the N-substituted heteroarenes is activated and reacts with aryl iodides to form a C-Se bond. The region of the C-H activation reaction is affected by the functional groups in the substrate [73], such as carbonyl, cyano, halogen substituent groups (mainly fluorine or chlorine), sulfur, nitrogen, and oxygen in the ring. C-H bonds around the nitro groups are more prone to activation reaction, so nitro groups will affect the position of activated C-H bonds and various asymmetric cyclic aryl selenide products can be obtained by activating C-H bonds at specific positions, which expands the selection range of substrates. However, such reactions usually require the use of expensive and possibly toxic metal salts or their ligands, harsh reaction conditions, and high reaction temperatures, so their application has certain limitations [74].

To address these shortcomings, researchers have developed a variety of selenation systems without transition metal catalysts, such as the application of ionic liquids (e.g., SeO_2_) or the use of visible light to catalyze the reaction, and a typical transition metal-free selenation system is shown in. Kour et al. [75] used SeO_2_ as a source of selenium in a dimethyl sulfoxide (DMSO) solvent system. First, the cyclic scaffold of pyrazole and other heteroaromatic hydrocarbons was reacted to synthesize bicyclic compounds, and then the selenium atom was introduced into the methylene (−CH_2_−) part between the two pyrazoles; the selenium was inserted in the form of monoselenoether (C-Se-C) and iselenoether (C-Se-Se-C) to construct monoselenoether pyrazole and iselenoether pyrazole (Figure 9a). Such bis-pyrazole compounds bound to selenium are outstanding in terms of biological applicability [76]. Zhou et al. [77] synthesized 3-selenium spiro indoline compounds by the visible-light-induced spiro cyclization of indolinones and diselenides in an air environment at room temperature (Figure 9b). The method shows a high yield, does not rely on metal catalysts, and exhibits strong anticancer activity in in vitro anticancer activity tests. He et al. [78] added coumarin derivatives, diaryl (alkyl) diselenides, and iodine chloride to the Schlenk reaction tube and added DMSO to the reaction under nitrogen protection. Selenized coumarin derivatives were obtained by rapid column chromatography (Figure 9c). Selenides can achieve a high yield and have characteristics of anti-tumor activity, simple synthesis, and weak toxicity.

With the help of an electrochemical reaction, the derivatization modification method of selenium and cyclic compounds driven by visible light has made great progress and can be used to synthesize a variety of selenium-containing cyclic compounds and promote the industrial production and drug development of such selenium-containing compounds [79].

### 6.2. Selenated Lipids

Selenite acid triglycerides did not originally exist in nature and are a mixture of several synthetic selenium-containing lipids [80]. Some researchers have hydroxylated the double bonds of some unsaturated fatty acids by the reaction of sunflower seed oil triglycerides with potassium permanganate at a controlled temperature. Subsequently, amorphous selenium was used as a catalyst to react the obtained product with selenite acid dissolved in dioxane to prepare a selenized triglyceride mixture (Selol) [80]. Structural characterization by high-performance liquid chromatography–inductively coupled plasma mass spectrometry (HPLC-ICP-MS) and high-performance liquid chromatography–electrospray ionization tandem mass spectrometry (HPLC-ESI-MS/MS) [81] revealed that selenium in such selenium-containing triglyceride derivatives is in the oxidation state of +4 valence, and its structural formula is shown in Figure 10. Selol was identified as a novel antioxidant and anti-tumor agent that is non-toxic, highly bioavailable, and active [82]. Sonet et al. [83] compared Selol, selenite, and selenate for selenium assimilation in human prostate cancer cells, human embryonic kidney cells, and human prostate epithelial cells. The results showed that selenium from Selol was effectively embedded in selenoproteins in human cell lines, significantly increasing the biosynthesis and activity of selenoproteins in the cell lines, and that selenating triglycerides were more effective and did not negatively affect cell growth and metabolism compared to other selenium sources. Grosicka-Maciąg et al. [84] discussed the regulation of Selol on adhesion molecules in normal and tumor necrosis factor-α (TNF-α)-stimulated human microvascular endothelial cells. The results showed that Selol could inhibit the expression of intercellular adhesion molecules induced by TNF-α and promote the production of reactive oxygen species.

In addition, research indicates that Selol displays anti-cancer characteristics in various cancer cell lines, such as human leukemia cells, cervical cancer cells, and breast cancer cells, albeit with low activity [85]. Flis et al. [86] discovered that Selol disrupts redox regulation in androgen-dependent prostate cancer cells (LNCaP), reducing the difference in redox status between extracellular and intracellular environments and exhibiting a potential pathway in regulating tumor cell apoptosis.

However, the Novel Food and Food Allergens (NDA) panel is required to comment on the safety of selenite triglycerides added to food supplements for nutritional purposes as a selenium source and the bioavailability of selenium from this source, according to regulations. Meanwhile, the panel finds that the information provided on the selenite triglyceride composition usually could not reflect the complete characterization of the product. According to the characterization data, it is almost impossible to determine which chemical form of selenium is available throughout the body and whether it can enter the functional selenium pool to realize the physiological function of selenium. Therefore, since there is no evidence that selenite triglycerides are converted into known forms of selenium after intake and absorption, selenite triglycerides should be regarded as heterophytes with unknown properties in vivo [80]. Due to the inherent difficulties in experimental detection and characterization, how selenite is released from lipids and whether all selenotriglycerides possess the same release ability need further study [83]. Other studies have found that selenized triglycerides toxicity to the body depends on the route of administration. Parenterally administered Selol is non-toxic, while its toxicity increases sharply when administered orally, which may indicate that more harmful products could be formed during digestion [82]. Nevertheless, the current research shows that selenium-modified lipids indeed exhibit particular anticancer activity, which brings hope for the treatment of advanced cancer patients, and its research is still of great significance. Next, further study will be necessary to depict the molecular target and detailed antitumor mechanism of Selol, which will contribute to promoting its application in tumor intervention and treatment.

### 6.3. Selenium Nanoparticles (SeNPs)

The nanosizing of selenium refers to the process of reducing the size of selenium particles to the nanoscale range [87,88]. SeNPs have been found to be less toxic than both the inorganic and organic forms of selenium, with the smaller SeNPs showing greater bioactivity [89]. Usually, methods for synthesizing the SeNPs can be categorized into three routes based on principles and synthesis conditions, including chemical, physical, and biosynthesis [90], of which chemical reduction is the most commonly used method to synthesize SeNPs [88].

Inorganic selenium, such as SeO_2_, Na_2_SeO_3_, NaHSeO_3_, and others, is often used as the selenium source in the chemical synthesis process, which is then reduced to its monomeric form with a reducing agent. In this process, ascorbic acid, gallic acid, glutathione, lignin, and folic acid are the commonly used reducing agents. Additionally, to prevent the agglomeration of SeNPs and enhance their stability, along with their biological activity, a stabilizer such as a template or a surfactant is usually added during the reduction reaction. The most typical templates used are macromolecules, including proteins, polysaccharides, and quercetin, that safeguard and steer the production of SeNPs from start to finish throughout the reaction process. These templates participate in rapidly triggering the formation of SeNPs nuclei, restrict their clustering and growth, and control the size of the SeNPs (Figure 11) [90]. The main physical synthesis methods for SeNPs are photocatalysis, pulsed laser ablation, and electrochemical methods. Among them, the photocatalytic method refers to synthesizing SeNPs by light reaction, which generates hydrated electrons and hydroxyl radicals. Hydrated electrons are characterized by solid reducibility and reactivity, while hydroxyl radicals are strongly oxidizing and reactive, resulting in reactions to produce nanoparticles [91]. Although the physical synthesis method possesses the incomparable advantages of other ways, such as not being contaminated by other chemicals and fast preparation, the disadvantages are also apparent, such as the high equipment cost and poor stability [90]. The biosynthesis of selenium nanoparticles involves utilizing bacterial redox systems to transform high-valent selenium into low-valent selenium. This process leads to the creation of consistent and stable selenium nanoparticles in the presence of intracellular or extracellular proteins [92]. The biosynthetic method is relatively simple, environmentally friendly, and requires no special conditions. Bacteria, fungi, and plants, among others, have been employed to yield nano-selenium under gentle conditions [93].

Compared to elemental selenium, SeNPs offer benefits, including greater solubility and improved biological activities [96], such as antioxidant, anti-tumorigenic, anti-stress, and bacteriostatic properties. Zhang et al. [95] prepared a monodisperse and stable selenium nanoparticle formulation (marked as Tw-TMP-SeNP, the size of the nanoparticle is 50 nm) using inulin and the surfactant Tween 80. Physicochemical analyses showed that the inulin and Tween 80 acted to tightly encapsulate the SeNPs through the formation of C-O⋅⋅⋅Se bonds or O-H⋅⋅⋅Se bonds. Tw-TMP-SeNP treatment significantly inhibited the proliferation of cancer cells in vitro, with HepG2 cells being the most susceptible, and their in vivo zebrafish model confirmed the anti-tumor activity of Tw-TMP-SeNP by inhibiting tumor cell proliferation, migration, and angiogenesis. Tang et al. [97] synthesized selenium nanoparticles (LAG-SeNPs) using arabinogalactan (LAG) as a scaffold and particle stabilizer to study their anti-tumor properties. Cytotoxicity assays showed that LAG-SeNPs had significant inhibitory effects on cancer cells, induced apoptosis, promoted shrinkage, and inhibited cell proliferation. LAG-SeNPs can be used as potential anti-cancer additives or drugs as they have the potential to inhibit cancer cell growth. Alvi et al. [98] synthesized metal nanoparticles from citrus fruit extracts using selenium metal salts and evaluated their antibacterial activity against pathogenic microorganisms. Antibacterial assays were carried out against Escherichia coli, Mycobacterium luteum, Bacillus subtilis, and Klebsiella pneumoniae and the results were compared with the standard antibiotic ciprofloxacin, which showed that the SeNPs showed comparable antimicrobial activity to the standard antibiotic ciprofloxacin. Significant antimicrobial activity was shown against all bacterial pathogens tested. Fourier transform infrared (FTIR) spectroscopy was used to detect functional groups involved in the synthesis of SeNPs, Indicating the presence of O-H bonding stretching, alcohol and phenol functional groups, and N-H stretching of amides.

Nano selenium shows a positive promise as a food supplement. It is commonly enclosed in polysaccharide carrier materials, including chitosan, glucomannan, gum arabic, or carboxymethyl cellulose. This method effectively delivers selenium to the cells, enhances selenium retention, bolsters the immune system, and reduces the risk of DNA damage. The research into selenium nanosizing has the potential to improve its bioavailability and safety, particularly for populations suffering from selenium deficiency and potential clinical applications [99].

## 7. Application and Prospect of Seleno-Modified Natural Products

### 7.1. Biomedical Field

When a selenized target is used as a drug, it can reduce its toxicity, enhance its biological activity, and also have the effect of selenium supplements. In addition, the target’s natural products can improve the body’s oxidative stress, enhance human immunity, and have a certain anti-tumor effect after selenization modification [100].

Zhang et al. [101] extracted glucomannan from the rhizome of Platycodon grandiflorum and modified it by the HNO_3_-Na_2_SeO_3_ method. The prepared selenized polysaccharides have a certain inhibitory effect on a variety of cancer cell lines, such as human lung adenocarcinoma cells, human breast cancer cells, and human gastric adenocarcinoma cells. At the same time, they show less toxicity to normal cells, which is expected to provide ideas for the development of new cancer treatment drugs. Sun et al. [102] confirmed that selenium-modified chitosan can effectively inhibit cancer cell proliferation, eventually leading to the apoptosis of hepatoma cell HepG2 by inducing mitochondrial dysfunction. In addition, Gao et al. [103] studied the regulatory effect of selenizing codonopsis pilosula polysaccharides (sCPPSs) on immune function in mice. The results of in vitro cell studies showed that sCPPSs could significantly promote the proliferation of mouse spleen cells and induce the secretion of immune factors, such as interleukin-2 (IL-2), while improving the phagocytic activity of mouse macrophages and the killing ability of natural killer cells. At the same time, the results of in vivo animal experiments showed that sCPPSs could significantly increase the serum total antibody level of mice and induce a specific immune response against ovarian protein antigen in mice, and sCPPSs are expected to become a new type of immune enhancer and immunomodulator.

Selenium-modified natural products have certain potential in improving nervous system diseases and intestinal flora ecology. Amporndanai et al. [104] used the parent nucleus benzoisoselenazolone of ebselen (a synthetic organic selenium drug with strong antioxidant and cytoprotective effects), which has a stabilizing effect on mutant superoxide dismutase-1, as a precursor to construct a series of organic selenium compounds. In in vitro and in vivo models, the neuroprotective effects of ebselen and its derivatives were studied. The results showed that ebselen and its derivatives revealed good neuroprotective effects in mouse nerve cells and restored the viability of mouse nerve cells in vitro. Wang et al. [105] showed that selenium-modified millet water-soluble dietary fiber has the ability to promote the production of tryptophan by intestinal flora. Tryptophan has a certain therapeutic effect on inflammatory gastrointestinal diseases, which helps promote the host’s intestinal health. Qin et al. [106] studied the effect of seleno-oligochitosan on intestinal dysfunction in piglets and found that seleno-oligochitosan could increase the activity of antioxidant enzymes, such as superoxide dismutase and glutathione peroxidase, enhance the antioxidant capacity of intestinal tissue, and reduce the damage of oxidative stress. In addition, seleno-oligochitosan promotes the growth of beneficial bacteria and reduces the breeding of harmful bacteria, thereby maintaining the balance of intestinal flora and promoting intestinal health.

### 7.2. Food Field

In food science, selenium-modified natural products can be used for food preservation, to improve food quality and taste, enhance food nutrition, and contribute to the development of functional foods, with a wide range of application prospects [107].

First, it can be used as a dietary supplement to improve the nutritional value of food [108]. For example, commercially available selenate cereals are made by adding selenium-modified natural products to grains to increase the selenium content of these grains [108], thereby helping to increase the human body’s intake of trace element selenium. Selenized food also has an antioxidant effect, which can reduce the risk of cancer to a certain extent with satiety [109]. Other researchers believe that selenium yeast [110] can be added to foods such as bread and milk. This form of selenium is more easily absorbed by the human body, effectively increasing the selenium content of foods and improving their nutritional value [111].

In addition, selenium-modified natural products with antioxidant and antibacterial activities have potential applications in the field of taste improvement and preservation [112]. For example, adding selenoproteins to meat products can improve the tenderness of food and its taste and prevent the oxidation and deterioration of meat products, prolonging their shelf life [113]. The main reason for this is that after selenium modification, these active natural products, as effective antioxidants, can inhibit the generation of free radicals in food, thus contributing to food preservation. Moreover, compared with traditional synthetic antioxidants, natural products often have better safety and stability after selenization modification [114].

In general, the selenization modification applied in natural products is an effective food improvement technology that can be used to elevate food quality and develop new functional foods in the future. It has broad application prospects in food science research.

### 7.3. Beauty and Skin Care Field

Adding selenium-modified active natural products to cosmetics can increase the antioxidant and antibacterial properties of products, helping skin resist the invasion of free radicals and effectively preventing skin problems caused by bacterial growth. Atopic dermatitis (AD) is a common skin disease characterized by barrier dysfunction, skin inflammation, and skin dysbacteriosis. Baldwin et al. [115] found that the use of hot spring water and moisturizer containing selenized products could manage inflammatory skin diseases. Moisturizer containing selenium-modified products can promote the growth of symbiotic bacteria, improve the microbial population structure, reduce the barrier function of skin and the symptoms of AD, and promote the recovery of skin homeostasis. Wei et al. [116] prepared selenium-enriched mung bean fermentation broth (Se-MBFB) by fermentation and carried out in vitro experiments. The results showed that Se-MBFB could enhance its physiological function with selenium. Se-MBFB is rich in polyphenols, peptides, and γ-aminobutyric acid. It has significant free radical scavenging and tyrosinase inhibitory activity, reduces melanin synthesis and deposition, and has a good whitening effect. At the same time, it can promote the production of collagen and cell regeneration, improve the elasticity and smoothness of skin, and has significant moisturizing and anti-aging effects. In addition, the selenized natural products in the fermentation broth also alleviate skin problems such as erythema, stains, and wrinkles.

## 8. Discussion

Selenium is a non-metallic element with an electronic configuration similar to sulfur. However, selenium usually exhibits a stronger polarization property than sulfur due to its larger atomic size and lower electronegativity, which resulted in a stronger reactivity and sensitivity of the selenium. Many natural products such as polysaccharides, proteins, and polyphenols contain functional groups with high reactivity, mainly including some nucleophilic sites such as hydroxyl, sulfhydryl, and amino groups. During selenization modification, the electrophilic electrons of the selenium atom may become attracted to the lone pair electrons present in the hydroxyl group. Therefore, the selenium element in various selenium sources can easily react with the hydroxyl and carbonyl groups in the natural product molecules to form selenium ester bonds, selenium oxygen double bonds, etc. In addition, selenium will undergo a displacement reaction with the sulfhydryl group on the natural product molecule, replacing it with a selenium-containing functional group to achieve selenization modification. Therefore, almost all natural products can be selenized to a certain extent, and the selenization method is still being explored and improved. In the process of selenization modification, the yield of products is an important index. According to the literature, the efficiency of various selenium modification methods for the distinctive natural products differed significantly. For example, in the field of selenized modification of natural products, selenized polysaccharide is one of the most commonly studied, and a high selenization efficiency (selenium content of the product was 38.27 mg/g) can be obtained when the sodium selenite method was applied, while the selenium content of seleno-sulfinic acid-modified protein could be as high as 75.85 mg/g and 69–90% of the substrate would be selenized when the polyphenols are modified by potassium selenocyanate. In addition, in terms of application, the selenization modification of natural products has shown a wide range of application prospects in many fields. In biomedicine, selenium-modified natural products can be used for drug research and development as they enhance the natural products’ biological activity and the body’s antioxidant stress, immunity, and anti-tumor ability. In food, selenium-modified natural products are used for food preservation, improving food quality and taste, and enhancing the nutritional value of food. In the field of beauty and skin care, selenium-modified active natural products are used in cosmetics. These products have antioxidant and antibacterial properties, aid in the skin’s resistance against free radicals and bacterial invasion, improve skin texture, and alleviate skin problems.

The selenylation modification of natural products is an emerging research field and one of the hot topics in current biochemical research. It has been widely used in medicine, food, and other fields, showing great prospects and application value. In general, there has been a broad and in-depth exploration and development of selenization methods and applications. The application prospect in the field of food science is particularly broad. In the future, it is necessary to further study the efficient selenization methods of different types of natural products and the internal molecular mechanisms of selenization modification to evaluate the safety and physiological activity of selenized products more comprehensively and to dig deeper into their application fields. With the continuous improvement and deepening of technical means and related scientific theories, it is believed that more efficient selenization modification methods will emerge, allowing the application potential of selenized natural products to be realized.

## 9. Conclusions

The changed chemical structure always results in an obviously, if not significant, different functional property than the selenized products. Numerous natural products with critical bioactivity have been successfully selenized for the purpose of developing functional food or specific medicines. This research established many effective ways to introduce selenium-containing functional groups or selenium into the target molecule. Currently, the mechanism of the selenization process and the bioactivity of the selenized products have been systematically studied, which lays a solid foundation for further exploring the functional properties of the natural products and their application fields. Future research would be attractive in exploring a more precise selenization mechanism for various natural products, as well as an in-depth evaluation of their bioactivity and safety.

## Figures and Tables

**Figure 1 foods-12-03773-f001:**
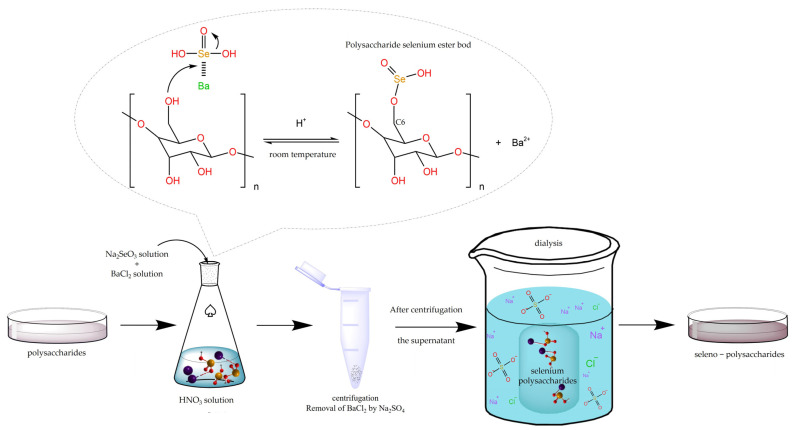
Process route of the HNO_3_-Na_2_SeO_3_ method [35].

**Figure 2 foods-12-03773-f002:**
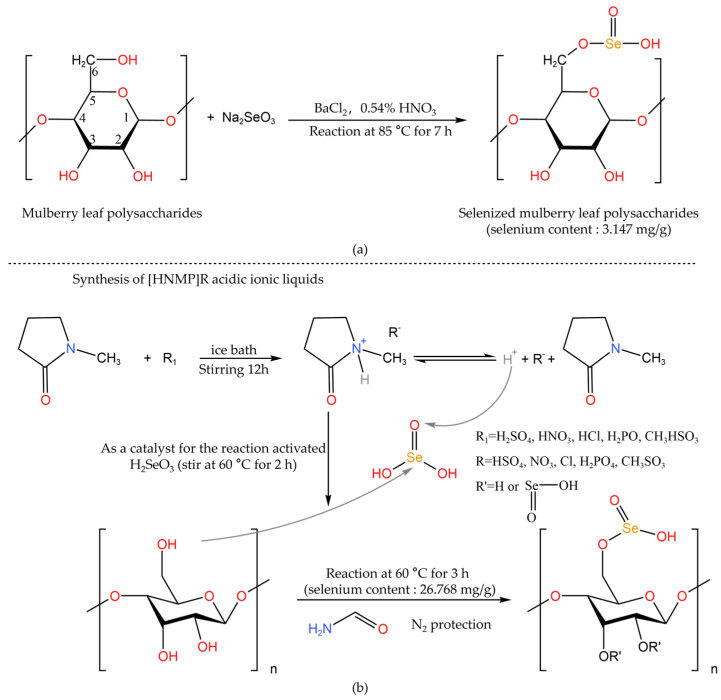
Selenium modification of polysaccharides [27,36]. (**a**) Reaction equation of selenization of mulberry leaves polysaccharides; (**b**) Synthesis of poly-saccharide selenate catalyzed by acidic ionic liquids.

**Figure 3 foods-12-03773-f003:**
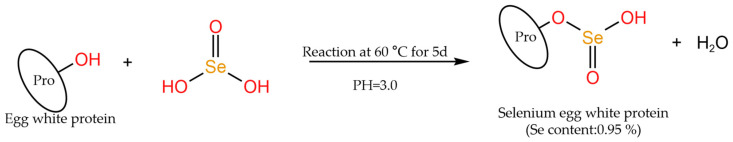
Selenylation of proteins to form a selenous ester bond.

**Figure 4 foods-12-03773-f004:**
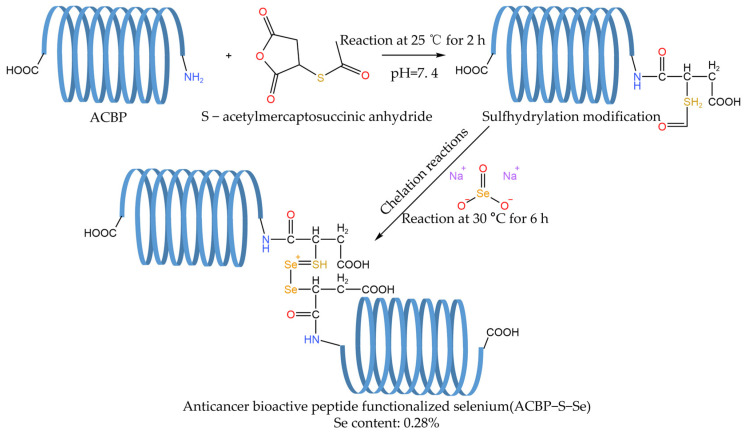
Synthesis of ACBP-S-Se.

**Figure 5 foods-12-03773-f005:**
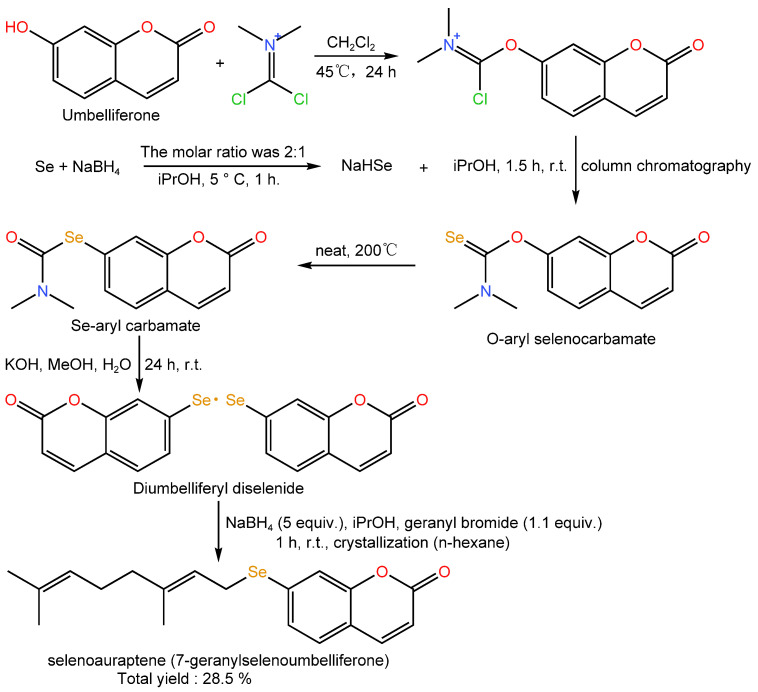
Semisynthesis of selenoauraptene.

**Figure 6 foods-12-03773-f006:**
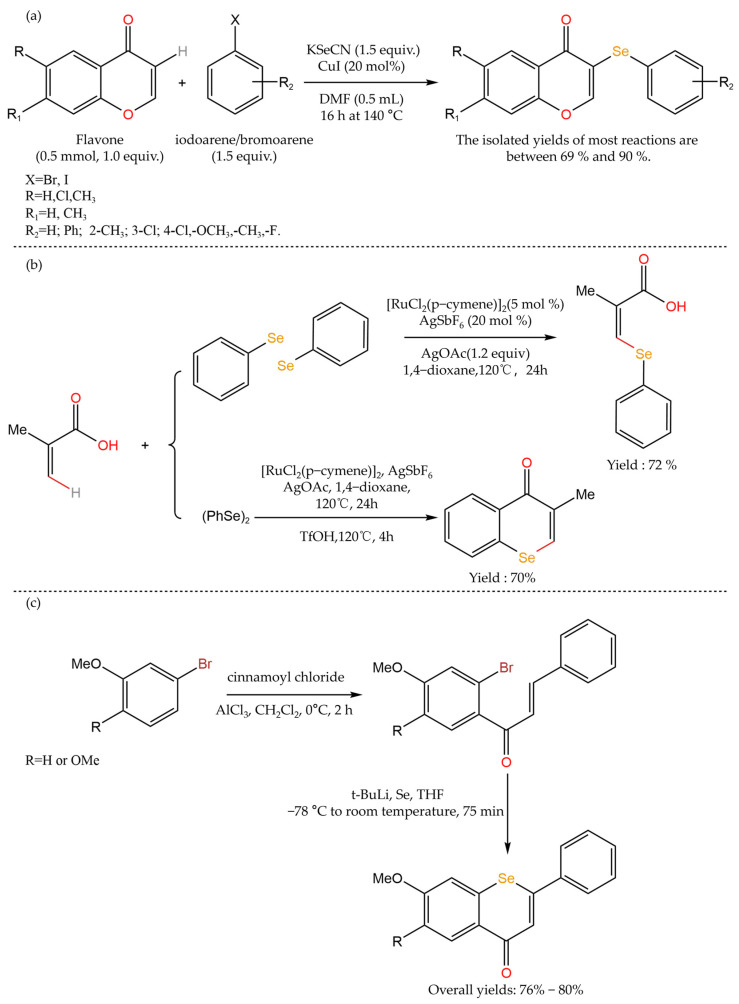
Selenium modification of polyphenols [63,64,65]. (**a**) Synthesis of selenium-containing flavonoid derivatives by different iodoaromatics/bromoaromatics and KSeCN; (**b**) Ruthenium-catalyzed synthesis of selenoflavones; (**c**) Two-step synthesis of flavanones selenide.

**Figure 7 foods-12-03773-f007:**
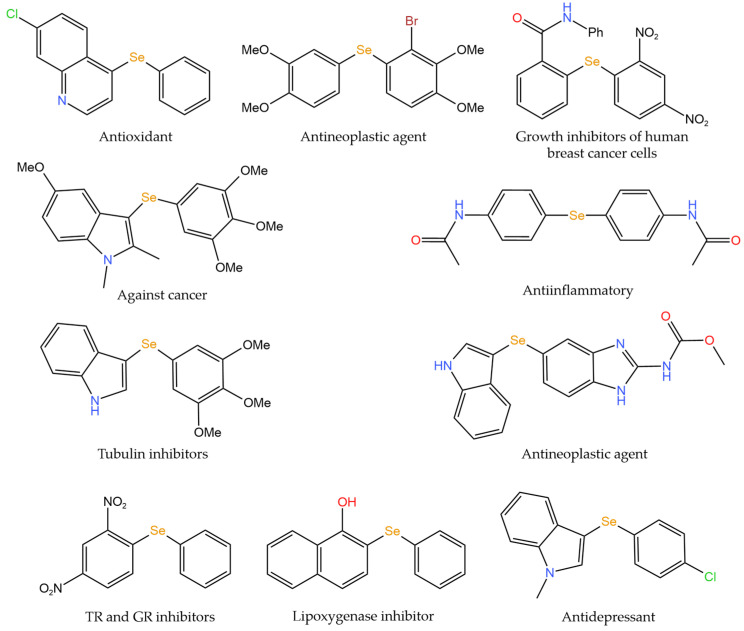
Biologically important aryl/heteroaryl selenides with wide activities [75].

**Figure 8 foods-12-03773-f008:**
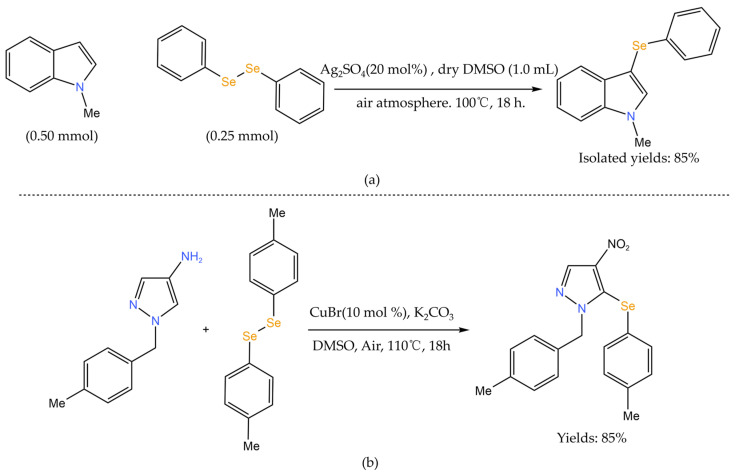
Selenium modification of cyclic compounds [70,72]. (**a**) Silver-catalyzed direct selenization of indoles; (**b**) Synthesis of 4-nitro-1H-pyrazole derivatives.

**Figure 9 foods-12-03773-f009:**
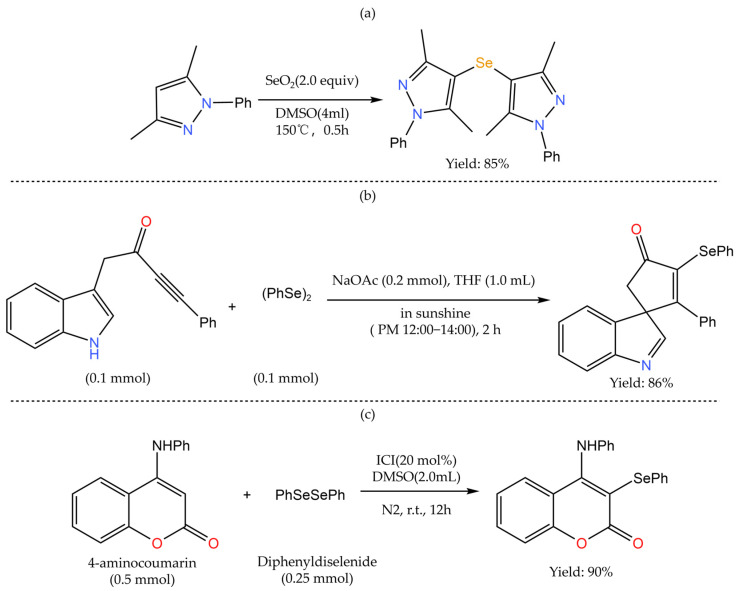
Selenium modification of heterocyclic compounds in the absence of transition metal catalysts. (**a**) The construction of selenylated bis-pyrazoles using SeO_2_ as the selenylating reagent; (**b**) Synthesis of seleno-spirocyclic compounds via selenylative dearomative cascade cyclization; (**c**) Synthesis of 3-arylselenocoumarin derivatives.

**Figure 10 foods-12-03773-f010:**
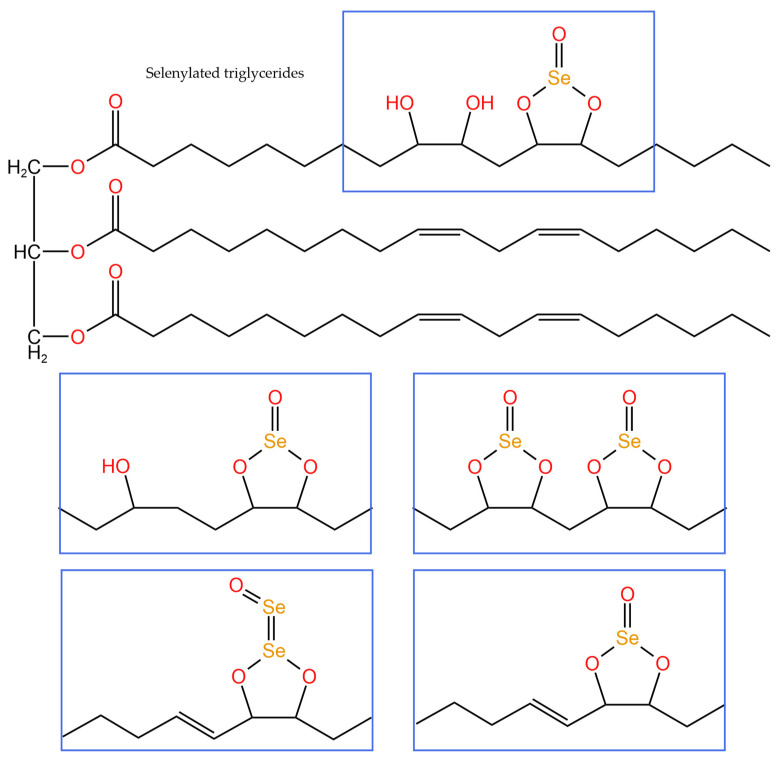
Chemical structure of the selenized triglycerides characterized by high-performance liquid chromatography–electrospray tandem mass spectrometry [83].

**Figure 11 foods-12-03773-f011:**
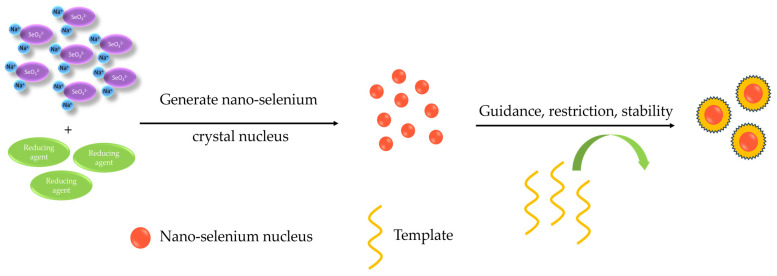
Schematic representation of the chemical synthesis of SeNPs [90,94,95].

**Table 1 foods-12-03773-t001:** Selenium modification method of polysaccharides.

Selenization System	Principal Operating Parameter	Selenium Content/(mg/g)	Combination	Reference
Aluminum chloride–sodium selenite (AlCl_3_-Na_2_SeO_3_)	(m(Na_2_SeO_3_):m(*Platycarya infructescence* polysaccharide)) = 1.60 The volume fraction of AlCl_3_ saturated aqueous solution is 4.56% microwave-assisted synthesis Reaction at 75 °C for 35 min	3.58	\	[29]
Nitric acid–sodium selenite (HNO_3_-Na_2_SeO_3_)	(m(Na_2_SeO_3_):m(garlic polysaccharide)) = 0.80 Reaction at 70 °C for 6 h pH = 5.5 Centrifugation at 3000 r/min for 10 min	38.27	Se-O-C Se=O	[30]
Glacial acetic acid–selenous acid (C_2_H_4_O_2_-H_2_SeO_3_)	(m(H_2_SeO_3_):m(garlic polysaccharide)) = 1.20 1% C_2_H_4_O_2_ solution 50 mL Reaction at 25 °C to complete dissolution Stir at 90 °C for 2 h 2 mL of 10% HNO_3_ solution was added and reacted for 8 h	26.3	C-O-Se Se=O	[31]
Nitric acid–selenous acid (HNO_3_-H_2_SeO_3_)	The polysaccharide was added to the HNO_3_ solution and stirred until completely dissolved Microwave power 190 W Stirring at 63 °C for 116 min Na_2_CO_3_ solution was adjusted to pH 6–7	2.6901	Se-O-C	[32]
Glacial acetic acid–sodium selenite (C_2_H_4_O_2_-Na_2_SeO_3_)	*Dendrobium devonianum Paxt.* polysaccharide: C_2_H_4_O_2_ = 1:1.5 (g/mL) *Dendrobium devonianum Paxt.* polysaccharide: Na_2_SeO_3_ = 1:1.5 (g/g) Water bath reaction at 60 °C for 24 h	12.37	O-Se-O Se=O	[28]
Selenium oxychloride (SOC)	500 mg polysaccharide was dissolved in 20 mL Fm solution. Stir at room temperature for 2 h SOC was dissolved by stirring at 60 °C for 3 h Reaction at 60 °C for 60 min	22.4	Se^4+^ replaced the C6 position	[23]
N-methyl-2-pyrrolidone hydrogen sulfate ([HNMP]HSO_4_)	(m(DMSO)):m(*Artemisia Sphaerocephala* polysaccharides)) = 0.10 Stirring at 80 °C for 5 h H_2_SeO_3_ was added to DMSO (40 mL) At 80 °C, [HNMP]HSO_4_ (equal molar ratio to H_2_SeO_3_) was added and stirred for 2 h The reaction was carried out under N_2_ for 6 h and centrifuged for 15 min Freeze-drying for 48 h	8.744	C-O-Se Se=O	[33]
Ascorbic acid–sodium selenite (C₆H₈O₆-Na_2_SeO_3_)	*Momordica charantia* polysaccharide powder (200 mg) dissolved in 200 mL of distilled water 10 mL *Momordica charantia* polysaccharide solution was mixed with Na_2_SeO_3_ solution (0.05 M) Move to 8 mL C₆H₈O₆ solution (0.1 M) oscillation at 28 °C for 12 h, dialyze with H_2_O for 48 h, freeze-dried	0.4354	C-O-Se Se=O	[34]

‘\’ means the information is not mentioned in the literature, similarly hereinafter.

**Table 2 foods-12-03773-t002:** Method of protein selenization modification.

Selenization System	Principal Operating Parameter	Selenium Content/(mg/g)	Combination	Reference
Microwave-assisted Na_2_SeO_3_ selenization	5 g Morchella protein hydrolysate, 20 g Na_2_SeO_3_ dissolved in 500 mL H_2_O pH = 5.0 adjusted by 1 M HCl Microwave power was 250 W, irradiation time was 20 min Centrifugation at 4000× *g* for 15 min	59	Se-O Se=O	[45]
CH_3_COOH-Na_2_SeO_3_	0.2 g *Nostoc* commune glycoprotein 0.2 g Na_2_SeO_3_, 0.5 g BaCl_2_ 50 mL 2% CH_3_COOH solution 2 mL 10% HNO_3_ solution The reaction was carried out in a 75 °C water bath for 7 h	1.403	\	[50]
Selenated sulfinic acid	10% selenated sulfinic acid solution buffer 6% H_2_O_2_ The reaction was carried out in a vacuum reactor at 40 °C for 5 h	75.85	Se=O	[51]
Dry-heat selenization	Egg white protein concentration of 2% The freeze-dried samples were kept at 60 °C for 5 d In the presence of H_2_^77^SeO_3_, pH = 3.0, dry heating at 60 °C for 3 d	9.5	-O-SeHO_2_	[46]
Na₂SeO₃	Selenized ovalbumin 2% 2% Na₂SeO₃ buffer solution pH = 3.0 The freeze-dried samples were kept at 60 °C for 3 d	8.8	S-Se-S	[49]

**Table 3 foods-12-03773-t003:** Methods for modifying polypeptides by selenization through chelation.

Selenization System	Principal Operating Parameter	Combination	Bioactivity	Reference
Na_2_SeO_3_-soybean protein isolate polypeptide	Na_2_SeO_3_:peptide = 1:2 (*v*/*v*) 0.1 mol/L Na_2_SeO_3_ solution Chelating at 80 °C for 2 h, pH = 7.0 Centrifuge at 6000 rpm and 4000 rpm for 10 min at 4 °C	−NH_2_, −NH, −COOH combined with Se	antioxidation	[6]
Na_2_SeO_3_-pea oligopeptides	Pea oligopeptide: Na_2_SeO_3_ mass ratio = 2:1 The final concentration is 5% Adjust the pH of the solution to 9.0 Chelating at 80 °C constant temperature H_2_O bath for 30 min	Se^4+^coordinated with −NH_2_ and −COOH	antioxidation	[55]
Na_2_SeO_3_-anticancer bioactive peptide	0.4 g S-acetyl mercapto succinic anhydride was incorporated into the anticancer bioactive peptide 2 h at 25 °C, pH = 7.4 N_2_ protection to avoid oxidation The reaction was carried out at 30 °C for 6 h with 0.66 g Na_2_SeO_3_	Se-Se, −C-Se, SH=Se	antioxidation anti-tumor	[56]

**Table 4 foods-12-03773-t004:** Method of selenium modification for polyphenols.

Selenization System	Principal Operating Parameter	Selenium Content/(mg/g)	Combination	Reference
Dihydromyricetin-Na_2_SeO_3_	2 g dihydromyricetin, 0.54 g Na_2_SeO_3_ 55% ethanol 60 mL 1 mol/L HCl to adjust pH to 3–4 Reaction at 55 °C for 20 min Vacuum drying for 48 h	6.54	C-Se	[66]
Tea polyphenols-selenium	0.2 g tea polyphenols were dissolved in H_2_O and added to the Se^4+^ solution. pH = 4–5 60 °C for 20 min	\	Se^4+^ substituted C-H on the benzene ring	[67]
Quercetin-SeO₂	Quercetin 0.75 g (2.5 mmol) 0.14 g (1.25 mmol) of SeO₂ dissolved in 10 mL of absolute ethanol. Keep the pH at 2–3 The reaction was carried out at 60 °C for 10 h under N_2_ protection	43.53	Se-O	[68]

## Data Availability

No new data were created or analyzed in this study. Data sharing is not applicable to this article.

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
