# Peer review of "Selenium Modification of Natural Products and Its Research Progress"

_foods, 2023, doi:10.3390/foods12203773_

Round 1
Reviewer 1 Report
The manuscript of Cheng et al entitled “Selenium modification of natural products and its research progress” reviews the development and application of selenization processes to modify natural products. The authors have discussed some examples of modifications of polysaccharides, proteins, peptides, polyphenols, lipids, and heterocyclic compounds. The manuscript has good technical quality with adequate language. As far as I researched, when addressing the insertion of selenium in compounds that could be used in food, the authors bring a relatively new perspective. However, I believe that some considerations could be made by the authors to improve the quality of the manuscript.
Major correction
1. In various parts of the manuscript, the authors discuss chemical reactions, but no scheme is presented. Thus, it may not be completely clear to the reader what is being discussed. Therefore, please, include more reaction schemes during the review. Even because, the central theme of the review, selenium modification (selenization) is a chemical reactions. Therefore, I consider this to be the main change that the authors need to make.
2. In “Discussion” the authors say “Many natural products … including some electrophilic sites such as hydroxyl, sulfhydryl, and amino groups.” Would not these cited sites be nucleophilic sites?
Minor correction
3. In the abstract, the authors have said “this paper forecasts the future development trend of the research field relating to selenized natural products…”. I suggest rethinking this sentence, as it does not seem to be very suitable. Maybe just trends, suggestions, indications...
4. Check if the abbreviation is correct in “selenium oxychloride (SOC)”.
5. In Figure 1, a chemical reaction is being represented, so the arrow cannot be curved. Furthermore, I suggest that authors use the generic chemical structures instead of the ball representation as it is. This would facilitate understanding, avoiding the need to place a legend with the colors of the atoms (balls).
6. In all schemes containing reactions (for example, in Figure 6) it is necessary to include the complete reaction information: solvent, reagents, time, temperature, and yield.
Reviewer 2 Report
This is an interesting review article in which authors have nicely summarized selenization of natural products. The manuscript is well written and various modes of synthesis as well as their biological applications have been well described. It would have been nice to include some selenium-based natural heterocycles in which selenium is inside the ring
1.
- M.A. Marć, A. Kincses, B. Rácz, M.J. Nasim, M. Sarfraz, C. Lázaro-Milla, E. Domínguez-Álvarez E, C. Jacob, G. Spengler, P. Almendros. Antimicrobial, Anticancer and Multidrug-Resistant Reversing Activity of Novel Oxygen-, Sulfur- and Selenoflavones and Bioisosteric Analogues. Pharmaceuticals. 13(12):453 (2020).
- 2. https://www.ncbi.nlm.nih.gov/pmc/articles/PMC7558951/
Overall it is nice manuscript and few minor comments are as under
1. please use italics for names of plants and expressions like "in vivo" and "in vitro"
2. if possible, only use, chemical formulae of chemicals in column 2 of of all tables
3. language issues:
abstract line 11, introducing
line 64, antioxidant instead of antioxidation
line 109 mL instead of ml
line 118, contamination instead of pollution
line 126-127. rephrase sentence "it was also observed...."
figure 3. please check if Na+ ion is needed
line 261. add NMR after nuclear magnetic resonance
page 11: monoselenoether and iselenoether instead of "Mono.." and "Dise..."
see the comments and suggestion section
Reviewer 3 Report
Review of the manuscript "Selenium modification of natural products and its research progress" written by
Kaixuan Cheng, Yang Sun, Bowen Liu, Jiajia Ming, Lulu Wang, Chenfeng Xu, Yuanyuan Xiao, Chi Zhang and
Longchen Shang. The manuscript is interesting and necessary, but the actual side of the matter needs to be corrected.
1. The authors need to clearly characterize the areas in the world, which are characterized by insufficient concentration of selenium in soils. The authors write that the public health problems caused by severe selenium deficiency are particularly prominent in China. And I do not agree with this statement in this form. For example, the soils of the southwest of Hubei Province are known to be selenium-excessive, as is, for example, part of the soils of Qinghai Province. In general, the authors need to formulate the proposal more carefully and list the main areas with selenium-deficient soils.
2. After the introduction, the authors immediately move on to selenium-saturated polysaccharides, claiming that such polysaccharides are very important. Such a transition introduces readers into bewilderment! The authors need to make a short chapter that would tell how selenium enters the body of mammals in nature (with which compounds, in which compounds).
3. The authors write that inorganic selenium can also combine with -OH, -NH2, -CHO, -C=O and other chemical groups on the polysaccharide chain. In this case, in inorganic selenium can form covalent bonds with the formation of selenite polysaccharide or selenate polysaccharide. Obviously, nothing like this happens under normal conditions. Authors need to indicate the specific conditions under which, in their opinion, such diverse chemical reactions can occur.
4. The text gives the impression that the selenization of protein molecules and polypeptide molecules is somewhat different. If the authors really think so, then they probably need to write what is the difference!
5. The manuscript has a section "5.1. Selenium-modified heterocyclic compounds", a significant part of the section and the figure are devoted not to heterocyclic compounds, but simply to cyclic ones. It is necessary to correct either the name of the section and the essence of the section.
6. The section of the manuscript "5.2. Selenated lipids" is highly controversial and debatable. I recommend either removing this section from the manuscript or providing information on the harm that non-canonical lipids cause in the human body.
7. The authors do not discuss selenization with selenium nanoparticles at all. This needs to be fixed! Perhaps selenium nanoparticles as a food additive and as a dosage form are the most common. Not long ago there was a very good work on this problem (10.3390/ma16155363). I recommend that authors use it when adding information to a manuscript. By the way, authors probably need to consider the antimicrobial properties of selenium and its compounds! This is important!
8. In the section of the manuscript "7. Discussion" the authors talk about the chemical properties of selenium. In my opinion, the authors greatly exaggerate the electrophilicity of selenium atoms, comparing and equating the properties of selenium with the properties of sulfur. In reality, selenium in aqueous solutions is often contained in the form of an oxide, which has different chemical properties. Free selenium is also not as active as the authors state... Probably, the discussion section needs to be substantially reworked. There is enough information in the manuscript to write the discussion in a more acceptable way.
Reviewer 4 Report
Manuscript “Selenium modification of natural products and its research progress” represents a contribution to field of fundamental and applied research in Foods sciences. Text is clear and easy to read. The research topic is original. The research topic presented in the manuscript is current. The literature used is adequate. Before accepting the manuscript, it is essential that the authors (make corrections):
In the discussion section, it is necessary:
- To quantify of efficiency (yield) of selenization, for each method separately. Compare the results and explain them.
- The purity of the selenides obtained is also important, especially for medical applications. It is necessary to quantify the purity of the selenides obtained for each method separately. Compare the results and explain them.
- It is necessary to add a conclusion to the manuscript.
Round 2
Reviewer 1 Report
After evaluating the changes made by the authors, I believe that the manuscript is suitable for publication.
Author Response
After evaluating the changes made by the authors, I believe that the manuscript is suitable for publication.
Authors reply:
We greatly appreciate the valuable feedback provided by the reviewer. The reviewer's suggestions were taken into serious consideration. your comments helped us improve the overall quality of our work. We are sincerely grateful for your guidance and support throughout the review process. Thank you once again for the insightful comments and for helping us enhance our manuscript.
Reviewer 3 Report
The authors corrected the incorrect information and the manuscript as a whole became better. However, there remains to be resolved one more important issue for readers.
The manuscript is devoted to the production of selenium-rich natural products. Currently, there are three directions for adding selenium to food. 1. Selenization - this direction of work is covered in the manuscript in sufficient detail. 2. Addition of selenium-containing inorganic compounds - this direction is covered in passing, but illuminated. 3. Addition of selenium nanoparticles. This direction has been completely ignored. The authors believe that their review is not devoted to selenium in the zero-valent state. The question is why? Maybe then the manuscript should be called “Modification of natural products with non-zerovalent selenium and the progress of its research”? I suggest that the authors take a different route and write a short chapter about nanoparticles. At least to me, such a chapter seems very important and interesting.
Author Response
The manuscript is devoted to the production of selenium-rich natural products. Currently, there are three directions for adding selenium to food. 1. Selenization - this direction of work is covered in the manuscript in sufficient detail. 2. Addition of selenium-containing inorganic compounds - this direction is covered in passing, but illuminated. 3. Addition of selenium nanoparticles. This direction has been completely ignored. The authors believe that their review is not devoted to selenium in the zero-valent state. The question is why? Maybe then the manuscript should be called "Modification of natural products with non-zerovalent selenium and the progress of its research"? I suggest that the authors take a different route and write a short chapter about nanoparticles. At least to me, such a chapter seems very important and interesting.
Authors reply:
Thank you very much for your review and valuable comments on our manuscript. We sincerely appreciate your attention and suggestions.
Upon further reflection, we recognized our limitations in organizing the paper and discussing the relative research topic. The caveats you raised about selenium nanoparticles are very reasonable, and we should indeed provide a relevant discussion of selenium nanoparticles in the manuscript.
We have added a chapter in the relevant section focusing on the research progress on preparation methods, biological activities, etc., of selenium nanoparticles. In this way, our manuscript will cover different methods of selenium addition in a more comprehensive way to meet the needs of our readers. Thank you again for your review and suggestions, which significantly improved the quality of the present manuscript.
Reviewer 4 Report
The authors corrected the manuscript in accordance with the reviewer's suggestions (to a large extent).
Author Response
The authors corrected the manuscript in accordance with the reviewer's suggestions (to a large extent).
Authors reply:
Thank you very much for taking the time to review our manuscript and providing valuable suggestions and comments. We have gladly incorporated your suggestions to a large extent and made significant revisions accordingly.
Once again, we greatly appreciate the insights provided in your review. Your suggestions are highly valuable to our research. We sincerely thank you for your professional input.